# A data-driven approach for real-time soft tissue deformation prediction using nonlinear presurgical simulations

**Haolin Liu**[ID][☯], **Ye Han**[☯], **Daniel Emerson**[ID], **Yoed Rabin**, **Levent Burak Kara**[ID]*

Department of Mechanical Engineering, Carnegie Mellon University, Pittsburgh, Pennsylvania, United States of America

☯ These authors contributed equally to this work.

* lkara@cmu.edu

**Data availability statement:** The code used in this work can be accessed at: https://github.com/HaolinCMU/softtissueAEtrack.

## Abstract

A method that allows a fast and accurate registration of digital tissue models obtained during preoperative, diagnostic imaging with those captured intraoperatively using lower-fidelity ultrasound imaging techniques is presented. Minimally invasive surgeries are often planned using preoperative, high-fidelity medical imaging techniques such as MRI and CT imaging. While these techniques allow clinicians to obtain detailed 3D models of the surgical region of interest (ROI), various factors such as physical changes to the tissue, changes in the body's configuration, or apparatus used during the surgery may cause large, non-linear deformations of the ROI. Such deformations of the tissue can result in a severe mismatch between the preoperatively obtained 3D model and the real-time image data acquired during surgery, potentially compromising surgical success. To overcome this challenge, this work presents a new approach for predicting intraoperative soft tissue deformations. The approach works by simply tracking the displacements of a handful of fiducial markers or analogous biological features embedded in the tissue, and produces a 3D deformed version of the high-fidelity ROI model that registers accurately with the intraoperative data. In an offline setting, we use the finite element method to generate deformation fields given various boundary conditions that mimic the realistic environment of soft tissues during a surgery. To reduce the dimensionality of the 3D deformation field involving thousands of degrees of freedom, we use an autoencoder neural network to encode each computed deformation field into a short latent space representation, such that a neural network can accurately map the fiducial marker displacements to the latent space. Our computational tests on a head and neck tumor, a kidney, and an aorta model show prediction errors as small as 0.5 mm. Considering that the typical resolution of interventional ultrasound is around 1 mm and each prediction takes less than 0.5 s, the proposed approach has the potential to be clinically relevant for an accurate tracking of soft tissue deformations during image-guided surgeries.

**Funding:** The author(s) received no specific funding for this work.

## Introduction

A major challenge that often hinders the success of image-guided surgery is the shape mismatch caused by the deformation of the region of interest (ROI), such as a tumor, between the time of diagnosis and the subsequent surgical procedure. Since biological tissues may be highly compliant, large deformations preceding or during the surgery may significantly reduce tissue tracking accuracy, leading to a critical mismatch between the real tissue configuration versus the imaging data presented to the clinician. Such discrepancies between the ROIs acquired during diagnosis and presurgical planning (commonly using MRI and CT imaging), versus those monitored during the surgery (CT or ultrasound) may arise due to a number of factors including physiological changes to the tissue over time, and differences in the patient's tissue configuration due to variations in the patient's 3D pose and contact with surgical instruments. Therefore, for surgeries that rely on 3D models acquired preoperatively, an accurate tracking and registration of the ROI deformation during the surgery is critical, especially for cases that require a precise control of safety margins such as physical tumor resection, cryosurgery, and radiation therapy [1].

To aid in downstream image-guided localizations and tracking of ROIs, fiducial markers (FM) [2–5] or functionally analogous markerless keypoint registration techniques [6–8], are commonly employed. FMs allow for clinicians to correlate between two images taken via different imaging modalities. In a typical radiotherapy application, for instance, FMs are implanted on the tumor or the surrounding tissue, which allows anatomical information of the ROI to be captured with a high resolution imaging modality such as MRI or CT. These markers are then easily detected during the surgery using lower fidelity systems such as ultrasound imaging, thereby allowing clinicians to track a tumor's general configuration in real-time. However, unless digitally registered with the preoperative high-fidelity model, a precise visualization of the ROI's instantaneous 3D shape can be very challenging, which possibly compromises the success of subsequent operations.

To account for the shape mismatch between surgical preplanning and intraoperative observations, researchers have made extensive efforts to explore different methods for tracking tissue deformations. Deformable models [9] have been proposed to simulate the shape change of soft biological tissues, including finite element methods (FEM). However, despite its accuracy, FEM simulations can be computationally prohibitive, especially when the solution involves complex mesh topologies and nonlinearities in either the computational models or material property formulations [10–20]. Even when tissue geometry can be rapidly captured via scanning, it is challenging for FEM to simulate its deformation in real-time for intraoperative visualization due to the need for computationally demanding, nonlinear iterative solvers. Several computational methods have been proposed including geometry and physics-based models. However, due to the intricate and unique nature of soft tissue boundary conditions, it is practically infeasible to create customized models for each tissue ROI and its interfacial environment to model the boundary conditions. Hence, there is an unmet need for surrogate models that can quickly and accurately predict tissue deformations using a limited amount of tracking aids such as the aforementioned embedded FM or keypoint registration techniques.

Recent literature has focused on combining machine learning (ML) methods with FEM to yield real-time tracking capabilities [21–25]. Some approaches use standard machine learning techniques for deformation prediction: Tonutti et al. use basic artificial neural networks and support vector regression [25], while Lorente et al. and Martinez et al. use tree based methods [21,24]. Karami et al. and Nguyen et al. used deep learning approaches built on LSTM methods to predict time varying deformations. Interestingly, Nguyen et al. use PCA to simplify the dimensionality of the deformation prediction as we do later in this paper. For clinical

adoption, we believe it is important that limited information on loading conditions and displacements is required to make prediction on the deformation of the ROI. All the aforementioned papers require information on the location, magnitude, or direction of the externally applied forces, which may be challenging to acquire intraoperatively.

In this work, we describe a data-driven approach capable of predicting nonlinear deformations of the surgical ROI based on a real-time, ultra-sound based tracking of a few FMs embedded in the tissue. A key advantage of our approach is that during the surgery, it does not require the knowledge of precise boundary or loading conditions surrounding the tissue. Instead, it only relies on the displacements of several embedded FMs, which are commonly tracked using ultra-sound imaging techniques. Given the patient-specific FM-embedded 3D tissue model extracted in a pre-surgery session, our approach first produces a rich dataset by simulating tissue deformations across a range of different boundary conditions. The FEM software Ansys is employed to build the simulation models, where we impose different boundary conditions (BCs) and regularization techniques to generate varying force distributions and corresponding 3D deformation fields. In the next step, we compute a concise latent space representation of each of the high-dimensional deformation field vectors using an autoencoder. Next, we train a neural network (NN) regressor that takes as input FM displacements, and outputs the latent space vectors of the deformation fields. Finally, the predicted latent space vectors are decoded to obtain the 3D deformation field vectors corresponding to the deformed tissue. Results show that given the displacement of as few as five FMs, our approach can reconstruct in real-time deformed tissues with sub-millimeter accuracy. We demonstrate our approach across a head and neck tumor, a kidney, and an aorta model.

## Materials and methods

Our proposed approach to predicting soft tissue deformations involves several major steps. Fig 1 details the overall pipeline, which can be further separated into three major components: (1) Establish FEM simulation models to generate various deformation fields given different boundary and loading conditions, thus creating a deformation dataset, (2) Train a convolutional autoencoder (AE) to encode the generated deformation fields into latent vectors, and (3) Train a neural network that maps simulated FM displacements to the corresponding latent vector of a particular deformation field. In clinical applications, the pretrained NN takes as input the observed FM displacements and outputs the latent vector. The latent vector is then decoded with the pretrained AE to reconstruct the deformed ROI.

To illustrate our approach, a 3D model of a head-and-neck (H&N) squamous carcinoma tumor is used throughout this section. The H&N tumor model is a retrospective de-identified model from a CT-scan of the tumor. The model was accessed for this study between January 23, 2018 and April 18, 2024. Researchers from this study had no access to identifying information over the course of the study, nor upon first receipt of the model. The CT-scanned H&N tumor is reconstructed using Synopsys' Simpleware™ software and meshed with a tetrahedral mesh in Netgen [26]. Fig 2 and Table 1 show further details of this geometry and its parameters. We also visualize the kidney and aorta aneurysm models used later in this paper, and detail their properties in Fig 2 and Table 1. It is worth noting that we used the same material properties from the H&N tumor for our kidney and aorta aneurysm models as well, since the major aim of this research is to show the capability of our data-driven pipeline on full-field deformation prediction given only a few displacements values of FMs, rather than generating realistic deformation fields for each respective model. We hypothesize that our trained machine learning model would yield equally accurate results with whatever material

## (a) Dataset Generation (Training Phase)

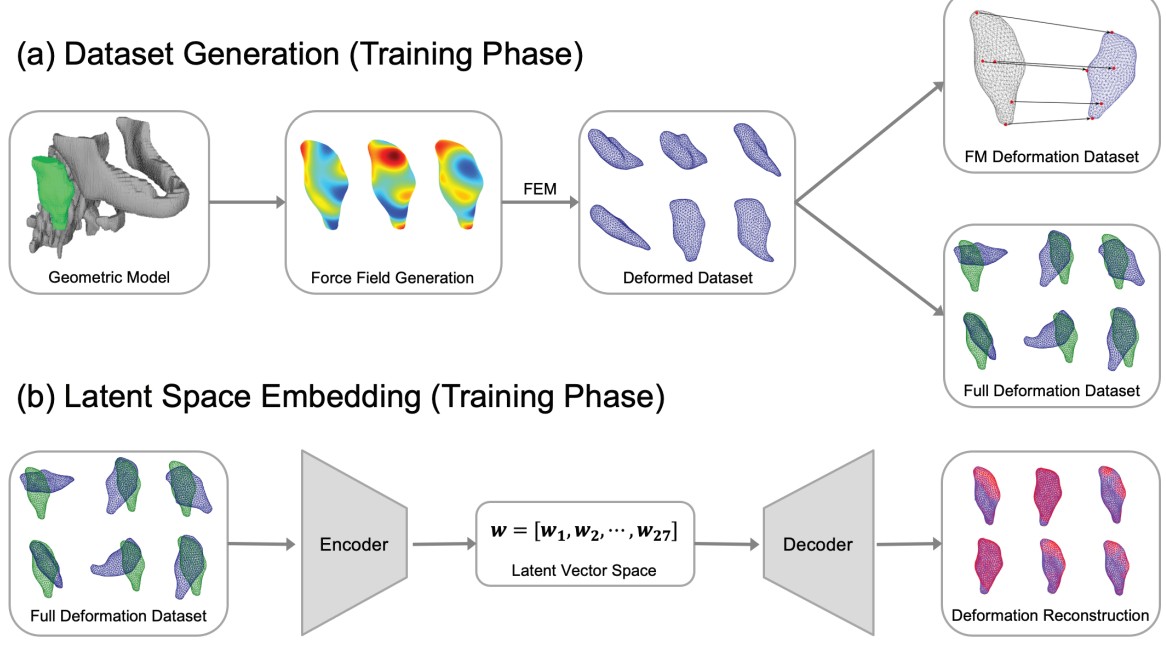

## (b) Latent Space Embedding (Training Phase)

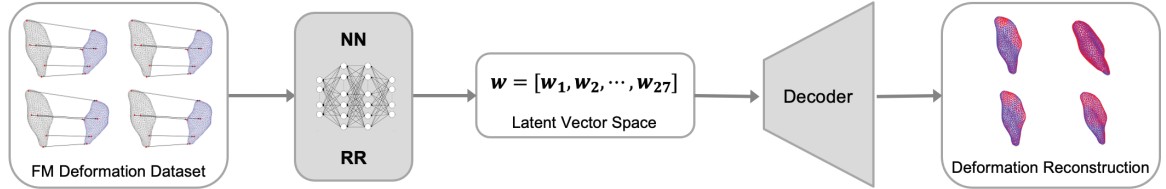

## (c) Deformation Regressor (Training Phase)

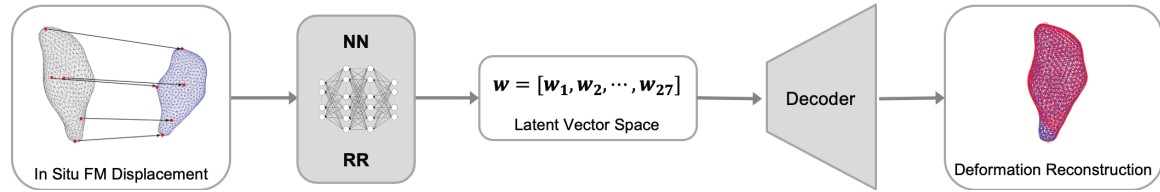

## (d) Deformation Prediction (Deployment Phase)

**Fig 1. Flowchart detailing training and deployment of the entire pipeline.** (a) Starting with a geometric model of the patient's region of interest (ROI), we create a deformation dataset with FEM from a widely varying force field. This dataset contains the displacements of all nodes in the model, but can also be used to track just the displacement of the fiducial markers (FMs). (b) Using the full deformation dataset, we train the autoencoder to create a latent space representation which encodes the shape deformations. (c) Also during the training phase, a neural network (NN) and ridge regression (RR) model are trained to map FM displacements to the latent space. (d) In the deployment phase, the NN or RR is used to map *in situ* FM displacements to the latent space representation. The trained decoder reconstructs the full shape deformation from the latent vector.

properties we choose as long as the soft tissue deformation magnitude falls within a similar range.

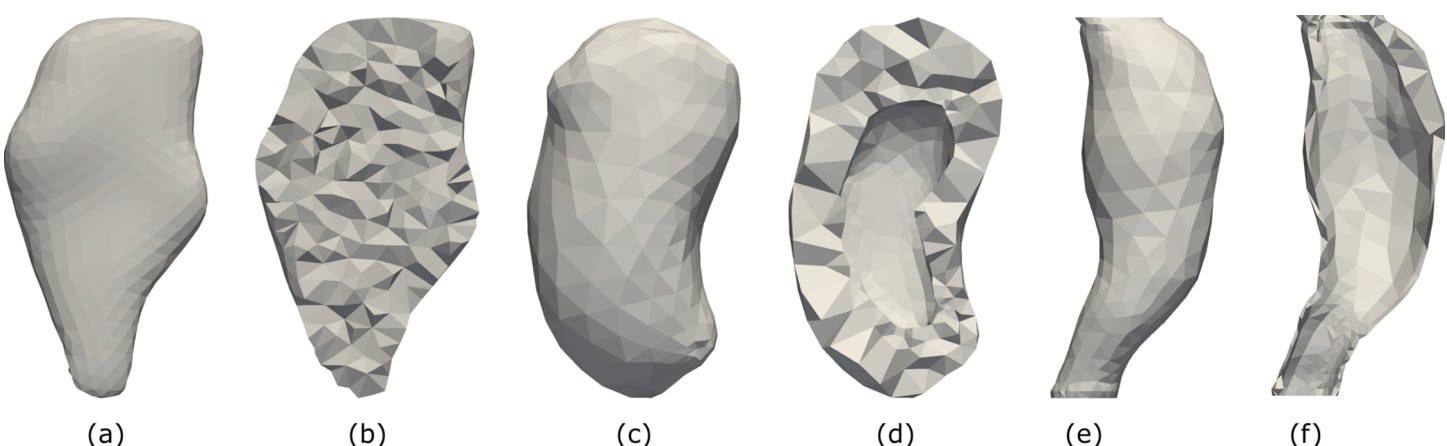

**Fig 2. CT reconstructed geometric models used in this study**. The models pictured are: (a) head and neck tumor, (b) head and neck tumor cross-section, (c) kidney, (d) kidney cross-section, (e) aorta aneurysm, and (e) aorta aneurysm cross-section.

**Table 1. Properties of the head and neck tumor, kidney, and aorta aneurysm models.**

| Property | Symbol | Head & Neck | Kidney | Aorta |
|---|---|---|---|---|
| Number of vertices | $n_v$ | 1158 | 1124 | 1218 |
| Number of elements | $n_e$ | 8520 | 4337 | 3518 |
| Number of surface vertices | $n_{sv}$ | 760 | 330 | 549 |
| Number of surface elements | $n_{se}$ | 1516 | 656 | 1044 |
| Maximum dimension (mm) | $d_{max}$ | 70 | 70 | 100 |
| Young's modulus (kPa) | $E$ | 21 | 21 | 21 |
| Poisson's ratio | $\nu$ | 0.45 | 0.45 | 0.45 |

## Modeling tissue deformation with nonlinear FEM

We use the nonlinear FEM solver in Ansys with quadratic tetrahedral elements for our simulation models. Within the FEM scheme [27], the tissue deformation is modeled as:

$$Kx = f \tag{1}$$

where $K \in \mathbb{R}^{3n_v \times 3n_v}$ ($n_v$ is the number of nodes) is the global stiffness matrix, $x \in \mathbb{R}^{3n_v \times 1}$ is the 3D nodal displacement vector and $f \in \mathbb{R}^{3n_v \times 1}$ is the external force vector. In this work, deformations are primarily driven by the forces generated in the tissue's interstitial environment. Hence, the external forces are assumed to be present only on the surface nodes. Thus, Eq 1 can be rewritten as:

$$\begin{bmatrix} K_{out} \\ K_{in} \end{bmatrix} x = \begin{bmatrix} f_{out} \\ 0 \end{bmatrix} \tag{2}$$

where $K_{in} \in \mathbb{R}^{3(n_v - n_{sv}) \times 3n_v}$ and $K_{out} \in \mathbb{R}^{3n_{sv} \times 3n_v}$ are submatrices of $K$ that correspond to the internal and surface nodes, respectively. As such, $f_{out} \in \mathbb{R}^{3n_{sv} \times 1}$ is the vector representing forces applied to the surface of the model.

With regard to the nonlinearity of the FEM solver, we model the following nonlinearities in the dataset generation:

1. **Material nonlinearity.** We model the soft tissue as a homogeneous, continuous and isotropic neo-Hookean solid. The material constants can be computed based on Young's modulus and Poisson's ratio following the approach introduced in [28].

2. **Geometric nonlinearity.** By separating the simulation process into incremental steps, the force is divided into load increments that are gradually applied to the tumor. The transformation and stiffness matrices are recomputed at each step to account for incremental changes in the geometry of the tumor.

3. **Nonlinear elements.** In our simulation, each element is a 10-node quadratic tetrahedral element with four corner nodes and six nodes located at the midpoints of the edges.

In essence, the nonlinearity of our simulation arises from both nonlinear material properties and nonlinear simulation schemes. This is sufficient to capture most of the nonlinear deformation behaviors of the target soft tissues.

## Creation of deformation dataset based on nonlinear FEM

To train our machine learning pipeline, we generate a dataset of deformed tumor models using the aforementioned nonlinear FEM solver. Since the deformation is primarily driven by a smooth force field on the tumor surface, we generate a large variety of unique smoothed force fields applied on the tumor surface and calculate the corresponding deformations through simulations. As shown in Fig 3, we first generate a few nodal forces on the surface of the model by randomly selecting positions, magnitudes, and directions of the loads; then we perform spectral decomposition of the discretized Laplacian $\mathcal{L} \in \mathbb{R}^{n_{sv} \times n_{sv}}$ of the tumor's surface mesh to obtain a discretized Laplacian-Beltrami operator (LBO) that can spread the nodal forces into smoother force field distributions around them. A scalar field $P$ of forces can then be reconstructed as a weighted linear combination of the first $k$ eigenvectors of $\mathcal{L}$ as follows:

$$P(x, y, z) = \Gamma \boldsymbol{w} \tag{3}$$

where $\Gamma$ is a $n_{sv} \times k$ matrix encoding the first $k$ eigenvectors of $\mathcal{L}$, $\boldsymbol{w}$ is a $k \times 1$ weight vector, and $P$ is the resulting scalar field defined over the surface of the tumor.

Using the above strategy, we can generate however many force fields $\boldsymbol{f}(x, y, z) \in \mathbb{R}^3$ we intend to, through a random sampling process of weight vector $\boldsymbol{w}$. As shown in Equ 4, to generate the $x$, $y$, and $z$ components of $\boldsymbol{f}$, we multiply those components with the aforementioned randomly generated weights and then reconstruct a series of completely different LBO-smoothed force field derived from a few concentrated forces ($f_{LSC}$).

$$\begin{bmatrix} | \\ F_j \\ | \end{bmatrix}_{n_{sv} \times 1} = s \begin{bmatrix} | & | & \cdots & | & \cdots & | \\ f_1 & f_2 & \cdots & f_i & \cdots & f_m \\ | & | & \cdots & | & \cdots & | \end{bmatrix}_{n_{sv} \times m} \begin{bmatrix} w_1 \\ w_2 \\ \cdots \\ w_i \\ \cdots \\ w_m \end{bmatrix}_{m \times 1} \tag{4}$$

where $j \in \{x, y, z\}$, $F_j$ is the $j^{\text{th}}$ component of $f_{LSC}$, $n_s$ is number of surface vertices, $s$ is the scaling factor, $f_i$ is eigen force field obtained from the $i^{\text{th}}$ eigenvector of the discretized LBO, $m$ is number of template force fields, and $w_i$ is weighting terms sampled uniformly from $[-1, 1]$. The scaling factor $s$ is used to control the magnitude of $f_{LSC}$, which subsequently determines the maximum displacement magnitude of the simulated deformations.

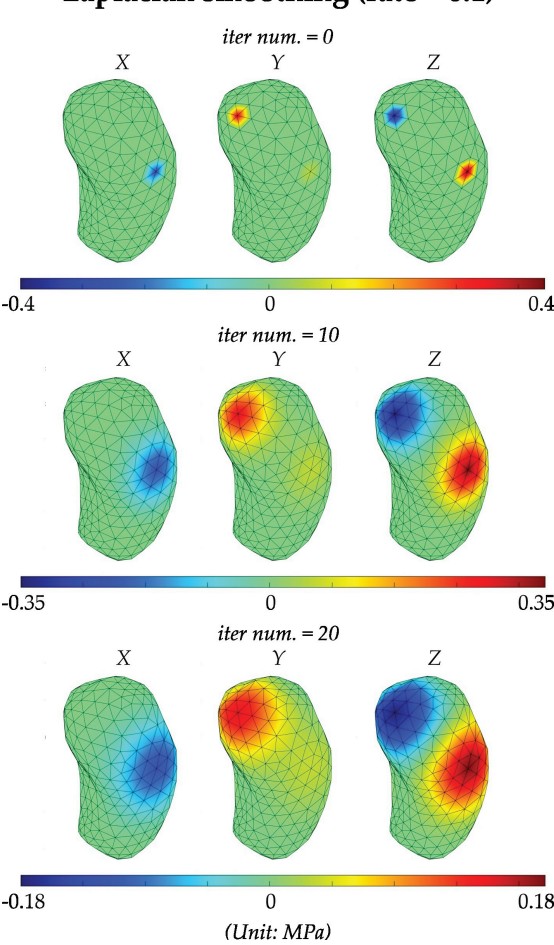

**Fig 3. Visualization of $f_{LSC}$ and intermediate force fields on the H&N tumor during the force Laplace smoothing process.** The procedures of generating $f_{LSC}$ are: (i) generate concentrated forces with a randomly sampled magnitude between $[-s, s]$ and a randomly assigned node on the outer surface as the loading position, and (ii) use LBO to locally smooth the concentrated forces and generate a smoothed concentrated force field ($f_{LSC}$). In the figure, the initial magnitude of three forces: $[f_x, f_y, f_z] = [5.0, 5.0, 5.0]$ N, and $f_{LSC}$ is eventually obtained after 20 iterations of smoothing with a rate of 0.1.

With the generated $f_{LSC}$, a pool of deformation samples can be simulated through the non-linear FEM simulation. The training $G_{train}$ and validation $G_{valid}$ dataset can therefore be partitioned and established from the generated deformation sample pool. To evaluate the prediction performance on much more diverse deformations, we create a set of randomly sampled $f_{LSC}$ and then simulate with given material properties and mesh structures. The detailed scheme of the construction of $f_{LSC}$ is shown in Fig 3. It is worth noting that we separately generate the testing dataset $G_{test}$ through nonlinear finite element simulation with completely different $f_{LSC}$, which will be used to validate the trained NN and evaluate the reconstruction performance.

## Encoding of deformation vectors

It is challenging to create a mapping from the displacement vector of a few FMs to the full-length deformation vector of a specific geometric model (e.g., a tumor), whose mesh contains

thousands of nodes and has very high dimensionality. Therefore, it would be impractical to put nodal displacements directly on the output side of the machine learning pipeline due to over-fitting and training time considerations. To overcome this issue, an AE is utilized to reduce the dimension of simulated deformation vectors and create a unique and concise latent representation for each deformation of a specific geometry. In this paper, a simple architecture with a stack of multiple convolutional layers is employed for both the encoder and decoder of the AE, with a $n$-dimensional latent space. On the application end, we train the AE immediately after we obtain the deformation results from the FEM simulations, and the resultant latent vectors are generated by pushing the full-length deformation vectors through the AE encoder. When we need to reconstruct the entire deformation vector from the corresponding latent embedding, we push the latent embedding through the trained AE decoder and the nodal displacements will be recovered accordingly.

## Deformation prediction with NN

After the AE is trained over the deformation field dataset, a NN is trained to predict the previously generated latent embeddings given the displacement vectors of FMs. We again utilize $G_{train}$ and $G_{valid}$, but this time we only consider the deformation of the FMs. We perform parametric studies (elaborated in Sec. Parametric studies) on input and output dimensions of the NN, corresponding to the number of FMs and latent dimension. We empirically found the hidden layer dimensions of the NN, $[n_1, n_2, n_3] = [1024, 512, 256]$, to balance generalizability with minimal overfitting across all geometries. Furthermore, we monitor the validation curve during training to ensure our model does not overfit to the training data. Our NN uses ReLU as the activation function for each layer except the output layer. The NN's structure and parameters (shown in Table 2) are the same for all geometries discussed in this paper.

In the proposed approach, the fully-trained NN serves as a surrogate model that projects the FMs' displacement vectors to the latent space established by AE. The predicted latent vectors are subsequently used to reconstruct the corresponding entire deformation field vector through the AE decoder.

## Deformation prediction with ridge regression

We use ridge regression as the baseline method for deformation prediction. The function parameters are detailed in Table 3 and the objective function is shown in Eq 5.

$$
\begin{aligned}
g(w_{RR}) &= \min_{w_{RR}} \left[ a_1 \left( \| D(P_{RR} w_{RR} + \bar{x}_{RR}) - d_{RR} \|_2^2 \right) + a_2 w_{RR}^T w_{RR} \right] \\
&= \min_{w_{RR}} \left[ a_1 \left( D(P_{RR} w_{RR} + \bar{x}_{RR}) - d_{RR} \right)^T \left( D(P_{RR} w_{RR} + \bar{x}_{RR}) - d_{RR} \right) \right. \\
&\qquad\left. + a_2 w_{RR}^T w_{RR} \right]
\end{aligned}
\tag{5}
$$

The objective function $g(w_{RR})$ in Eq 5 minimizes the difference between reconstructed and ground truth FM displacements $u_{RR} = D(P_{RR} w_{RR} + \bar{x}_{RR}) - d_{RR}$ with respect to the latent vector $w_{RR}$. The coefficients $a_1$ and $a_2$ are weighting terms with $a_1 >> a_2$. We apply a small weight, $a_2$, on the $L_2$ regularization term $w_{RR}^T w_{RR}$ to prevent overfitting. The optimal ratio of $a_2/a_1$ is obtained through a parametric study for each model.

## Evaluation metrics

To evaluate our models' ability to reconstruct the shape of the deformed tissue, we use the two following metrics: mean nodal reconstruction error (Eq 6) and max nodal reconstruction

**Table 2. Parameters of benchmark dataset and ANN structure.**

| Model Parameter | Symbol | Value |
|---|---|---|
| Number of training samples | $n_{train}$ | 2000 |
| Number of test samples | $n_{test}$ | 200 |
| Number of validation samples | $n_{val}$ | 200 |
| Number of neurons [layer$_1$, layer$_2$, layer$_3$] | $[n_1, n_2, n_3]$ | $[1024, 512, 256]$ |
| Number of FMs | $n_D$ | 5 |
| Maximum displacement in test samples (mm) | $x_{test\_max}$ | 30 |
| Training epochs | epoch$_{train}$ | 12000 |
| Batch size | $b_s$ | 128 |
| Learning rate | $\beta$ | 0.0001 |

**Table 3. Ridge regression parameters.**

| Parameter | Symbol | Dimension |
|---|---|---|
| Nodal displacements of a single deformation benchmark | $x_{RR}$ | $\mathbb{R}^{3n_v \times 1}$ |
| Mean nodal displacement across all benchmarks | $\bar{x}_{RR}$ | $\mathbb{R}^{3n_v \times 1}$ |
| Ground truth nodal displacement of $n_d$ FMs | $d_{RR}$ | $\mathbb{R}^{3n_d \times 1}$ |
| Binary indicator matrix, $Dx_{RR} = d_{RR}$ | $D$ | $\mathbb{R}^{3n_d \times 3n_v}$ |
| Principal components | $P_{RR}$ | $\mathbb{R}^{3n_v \times n_w}$ |
| Principal component reconstruction weights | $w_{RR}$ | $\mathbb{R}^{n_w \times 1}$ |

error (Eq 7),

$$\text{Offset}_{\text{mean}} = \frac{1}{n_v} \sum_i^{n_v} \left\| x_{bench}^i - x_{pred}^i \right\|_2 \tag{6}$$

$$\text{Offset}_{\text{max}} = \max_i \left\| x_{bench}^i - x_{pred}^i \right\|_2 \tag{7}$$

where $x_{bench}^i$ and $x_{pred}^i$ are $3 \times 1$ displacement vectors of vertex $i$ in the benchmark and predicted shapes and $n_v$ is the total number of vertices. In each case we compute the Euclidean distance between a given benchmark vertex and predicted vertex. For the mean nodal reconstruction error, we average the error over all vertices in a given shape. For the max nodal reconstruction error, we simply take the maximum error of all the vertices in the shape, to get an idea of the worst case error for our models.

## Results and discussion

In this section, we demonstrate the deformation reconstruction performance of our proposed approach on three different geometric models - namely H&N tumor, kidney, and aorta aneurysm. We also present the parametric study results for latent space dimension, number of FMs, and NN or AE architectures.

We implement the NN with PyTorch 1.6.0, and all demonstrated experiments are completed using a consumer-grade CPU (4-core Intel I7-8550U @ 1.80GHz). Each geometry is discretized as a mesh model, with the topology of each model remaining fixed throughout the FEM simulation process as well as the learning pipeline. Table 1 shows the fundamental parameters, including material constants, of each geometry and its corresponding mesh model. The number of FMs is prescribed before running through the machine learning pipeline, and the FMs are selected from the nodes of each mesh model following the revised k-center clustering algorithm (detailed in S1 Appendix). After the learning process, we validate the reconstruction performance on all geometries with both the fully-trained pipeline

and ridge regression (our baseline method), to demonstrate the efficacy and efficiency of our proposed method.

## Head-and-neck tumor

For the H&N tumor geometric model, we use displacement vectors of 5 FMs and latent vectors with the dimension of $12 \times 1$ as the input and output of the NN training, respectively. The corresponding full-length deformation field vectors are reconstructed using the pretrained AE decoder given the predicted latent vectors. To ensure diversity in the dataset we deform the ground truth FEM simulations up to a maximum magnitude of $\sim 70$ mm, for widely varying force field distributions as described in Sec. Creation of deformation dataset based on non-linear FEM and Fig 3. We detail the scale of deformations seen in the three models in Fig 4. Fig 6 visualizes the benchmark configuration and the reconstructed configuration of the H&N

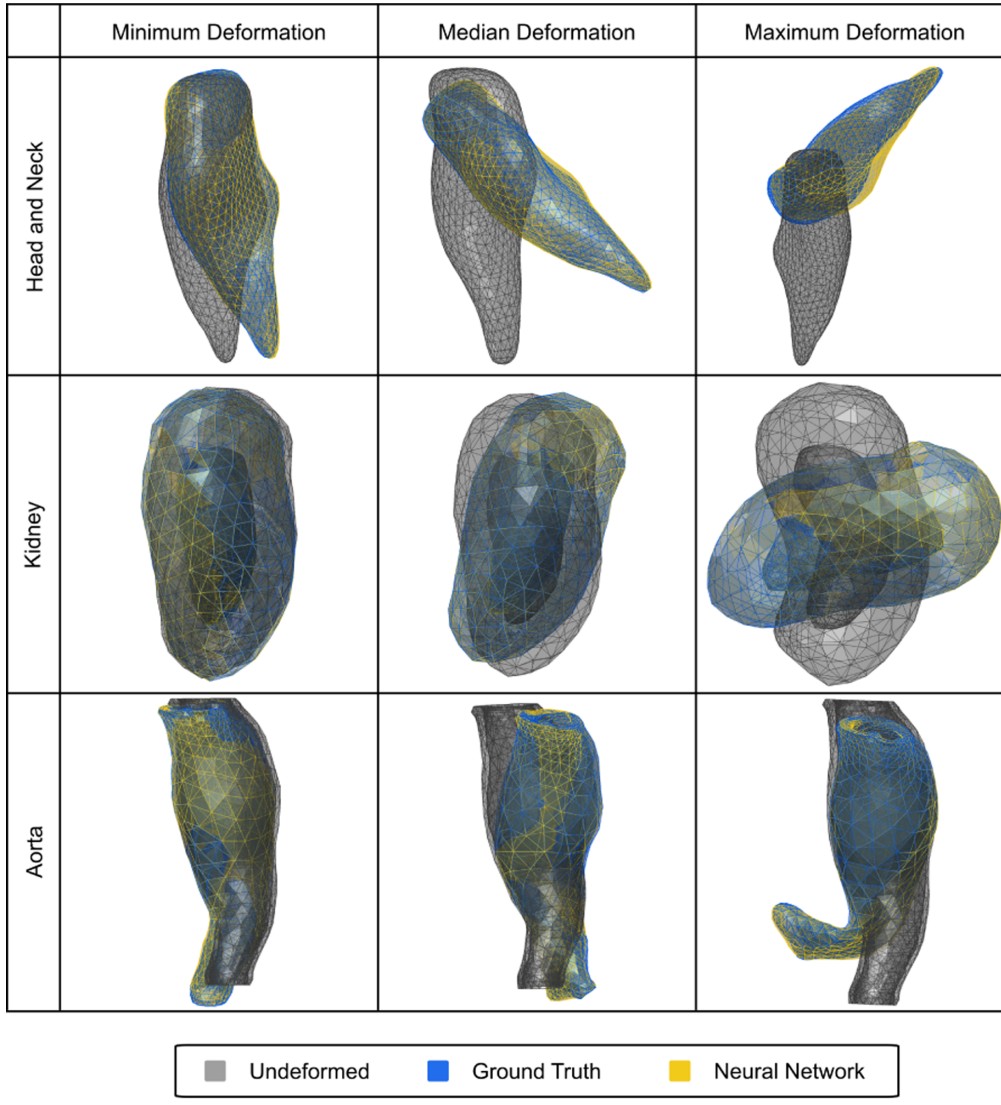

**Fig 4. Minimum, median, and maximum deformations for all three geometries along with undeformed geometry and NN reconstructed deformation.** The gray mesh corresponds to the undeformed geometry, the blue mesh corresponds to the ground truth deformation, and the yellow mesh corresponds to the NN reconstruction.

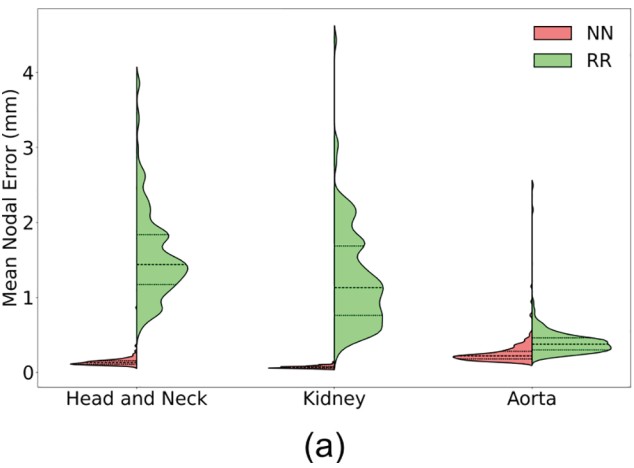
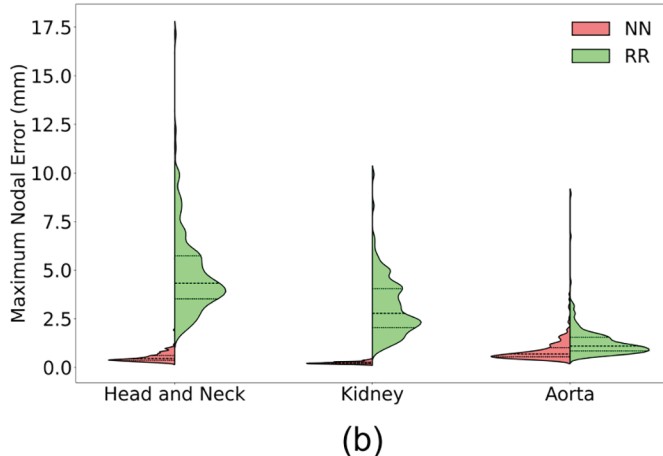

**Fig 5. Violin plots comparing reconstruction error between the NN and RR models on the test set for all three geometries.** Red corresponds to the NN model, green corresponds to the RR model, and quartiles are marked by the dashed lines. (a) Mean nodal reconstruction error. (b) Maximum nodal reconstruction error.

**Table 4. Mean nodal reconstruction error averaged across all train and test cases for the three geometries for NN and RR models.**

| Geometry | Train | | Test | |
|---|---|---|---|---|
| | NN (mm) | RR (mm) | NN (mm) | RR (mm) |
| Head and Neck | 0.100 | 1.504 | 0.141 | 1.553 |
| Kidney | 0.064 | 1.274 | 0.069 | 1.247 |
| Aorta | 0.174 | 0.405 | 0.248 | 0.410 |

**Table 5. Max nodal reconstruction error averaged across all train and test cases for the three geometries for NN and RR models.**

| Geometry | Train | | Test | |
|---|---|---|---|---|
| | NN (mm) | RR (mm) | NN (mm) | RR (mm) |
| Head and Neck | 0.387 | 4.654 | 0.514 | 4.842 |
| Kidney | 0.226 | 3.139 | 0.236 | 3.106 |
| Aorta | 0.616 | 1.305 | 0.859 | 1.318 |

tumor based on FM displacement samples from $G_{test}$ using both the NN and RR models. Fig 5 shows the violin plots of mean nodal reconstruction error (Offset$_{mean}$) and max nodal reconstruction error (Offset$_{max}$). The average reconstruction error values are tabulated for both NN and RR models across all geometries in Table 4 for Offset$_{mean}$ and in Table 5 for Offset$_{max}$. Importantly, the maximum nodal reconstruction error for test cases on all three geometries is sub-millimeter, meaning that our approach exceeds the resolution of conventional intraoperative ultrasound.

## Kidney

A kidney consists of two major parts, the medulla and the cortex [29]. In our implementation, it is assumed that the medulla has an extremely soft mechanical property so that the physical interaction between medulla and cortex can be ignored. It is common to model the kidney as a homogeneous material, with no distinction made between the medulla and cortex [30,31]. We study the cortex of the kidney, which has a hollow structure and is therefore

topologically different from the H&N tumor. For the kidney model, we use displacement vectors of 5 FMs and latent vectors with the dimension of $8 \times 1$ as the input and output of the NN training, respectively, and the maximum deformation magnitude is around 30 mm. Fig 6 visualizes the benchmark configuration and the reconstructed configuration of the kidney model based on FM displacement samples from $G_{test}$ using both the NN and RR models. Fig 5 show the violin plots of mean nodal reconstruction error ($Offset_{mean}$) and max nodal reconstruction error ($Offset_{max}$), respectively. The average reconstruction error values are tabulated for both NN and RR models across all geometries in Table 4 for $Offset_{mean}$ and in Table 5 for $Offset_{max}$. From the above results, it can be concluded that on the geometric model with a different topology, our proposed approach still has a higher reconstruction accuracy and a much faster reconstruction speed than a traditional baseline model like ridge regression.

## Aorta aneurysm

Different from the previous two models, the aorta aneurysm geometric model is a thin tube-like shape with a hollow structure [32]. Using the same approach, we apply force fields to the outer surface of the geometry, and all six degrees of freedom of the three pre-selected nodes at the middle of the geometry are fixed as the boundary conditions. Since the structure of the studied aorta aneurysm can be easily deformed with lateral forces (forces perpendicular to its

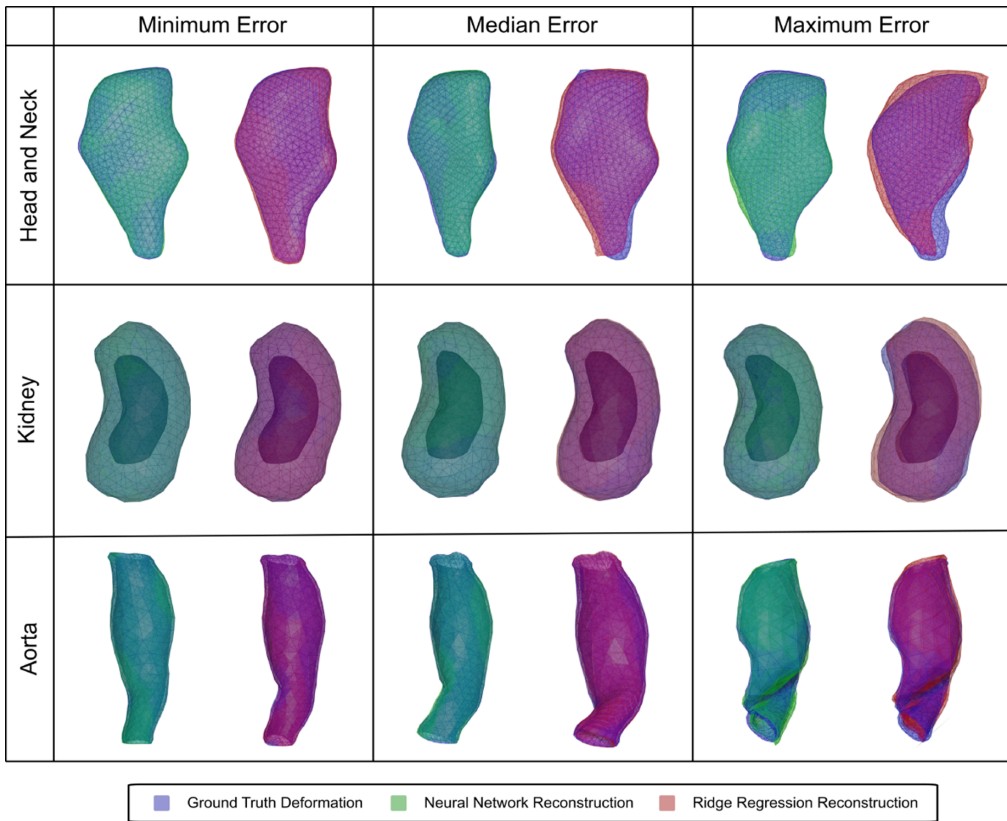

**Fig 6. Reconstructed deformations on all three geometries for NN and RR models, corresponding to the minimum, median, and maximum error cases between the reconstructed and ground truth deformations.** The blue mesh corresponds to the ground truth deformation, the green mesh corresponds to the NN reconstruction, and the red mesh corresponds to the RR reconstruction.

middle axis), we control the magnitude (smaller than the force fields applied to the above two models) of the general forces to make the maximum deflection of the model reach 30 mm.

For the aorta aneurysm geometric model, we use displacement vectors of 10 FMs and latent vectors with the dimension of $20 \times 1$ as the input and output of the NN training, respectively. Fig 6 shows the visualized comparison between the benchmark configuration and the reconstructed configuration of the aorta aneurysm geometric model. Fig 5 shows the violin plot of $Offset_{mean}$ and $Offset_{max}$ of two different reconstruction methods. On the model of aorta aneurysm, our proposed approach outperforms the baseline method again, with a lower mean and max nodal reconstruction errors on both the training and testing sets. This result, together with the reconstruction performance of the kidney geometric model, demonstrates the generalizability of our proposed approach to different geometries and topologies.

## Parametric studies

In this section, parametric studies are accordingly demonstrated with respect to the number of FMs and the dimension of the latent space. These correspond to the input and output dimensions of the neural network, respectively. We showcase the results using the H&N tumor. The optimal parameters are determined by not only having an acceptable reconstruction error defined in Sec. Evaluation metrics, but also considering the surgical and computational cost.

**Number of FMs.**   In clinical practice, FMs correspond to the pre-surgically implanted gold seeds/physical deformation trackers inside the human body. Therefore, it is desirable to have as few FMs as possible during surgery. In practical applications, there may also be a limited region where FMs can be placed. We perform a parametric study to analyze how prediction accuracy varies with the number of FMs used. In this section, experiments are performed with the number of FMs varying from 1 to 30. The FMs are placed using the k-center clustering algorithm described in S1 Appendix. The results of the parametric study are shown for the

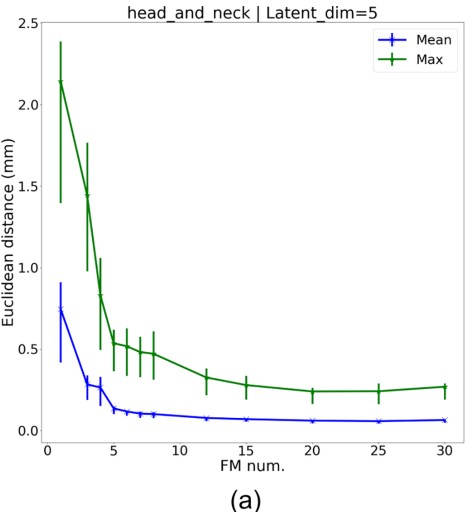
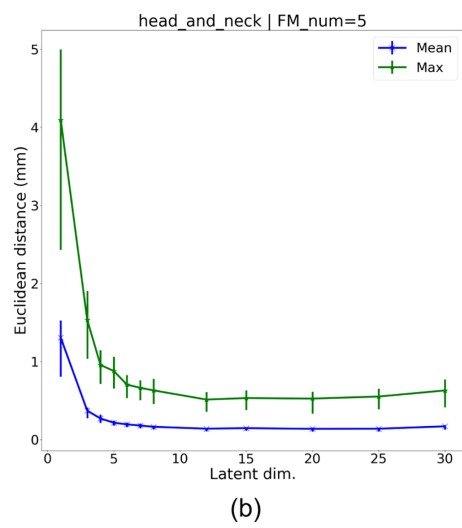

 (a) (b)

**Fig 7. Parametric study results for varying numbers of FMs and latent dimensions on the H&N tumor model.** The average nodal displacement errors (i.e., mean nodal error and max nodal error) are plotted together with error bars. (a) Varying number of FMs, with latent dimension of 5. (b) Varying latent dimension with 5 FMs.

H&N tumor in Fig. 7. We note that when the number of FMs decreases, both the overall prediction error and the number of significantly-poor cases increase. Likewise, for the other two investigated geometric models, similar correlations between the number of FMs and the prediction performance is observed. To select the optimal FM number for a specific geometric model, we set a threshold for $\text{Offset}_{mean}$ or $\text{Offset}_{max}$ and pick the fewest possible number of FMs that yields a result under this threshold.

**Latent dimension.** The latent dimension may vary with different geometries. It is apparent that the larger the latent dimension is, the more accurately the pretrained AE decoder can reconstruct a deformed configuration. Nevertheless, we should also apply a reasonable limit to the latent dimension to make the NN training process less challenging. To resolve

**Algorithm 1. Revised K-center clustering.**

**Input :** Node set V, initial node $v_0$, number of centers k, minimal distance threshold $d_{min}$
**Output:** The list of all searched centers $V_C$

```
 1  begin
 2      Initialize i ⟵ 0, V_C ⟵ ∅, S ⟵ ∅; Add v_0 ⟶ V_C;
 3      while i < k do
 4          Set maxAvg ⟵ 0, minVar ⟵ 1e5, v_new ⟵ ∅;
 5          foreach v_j ∈ V do
 6              if v_j ∈ V_C then
 7                  Skip to the next loop;
 8              else
 9                  S ⟵ ∅;
10                  foreach c_m ∈ V_C do
11                      Compute euclidean distance between v_j and c_m: d_m = norm(v_j, c_m);
12                      Add d_m ⟶ S;
13                  end
14                  Compute average of S: avg_j = avg(S);
15                  Compute minimum of S: min_j = min(S);
16                  if avg_j > maxAvg and min_j > d_min then
17                      v_new ⟵ v_j;
18                      maxAvg ⟵ avg_j;
19                  else
20                      Skip to the next loop;
21                  end
22              end
23          end
24          Add v_new ⟶ V_C; Set v_new ⟵ ∅;
25          foreach v_n ∈ V do
26              if v_n ∈ V_C then
27                  Skip to the next loop;
28              else
29                  S ⟵ ∅;
30                  foreach c_p ∈ V_C do
31                      Compute euclidean distance between v_n and c_p: d_p = norm(v_n, c_p);
32                      Add d_p ⟶ S;
33                  end
34                  Compute variance of S: var_n = var(S);
35                  Compute minimum of S: min_n = min(S);
36                  if var_n < minVar and min_n > d_min then
37                      v_new ⟵ v_n;
38                      minVar ⟵ var_n;
39                  else
40                      Skip to the next loop;
41                  end
42              end
43          end
44          Add v_new ⟶ V_C; i ⟵ i + 1;
45      end
46  end
```

this trade-off, we run parameterization on the latent dimension to search for the optimal one that takes both the computational cost and the reconstruction accuracy into consideration. Fig 7 shows the parameterization result on the latent dimension of the H&N tumor model. We identify an "elbow" in both the mean and max nodal displacements after the latent dimension reaches 5, and choose this as our latent dimension. We also observe that very high latent dimensions (i.e., close to 30 in this case) might lead to worse reconstruction performances. This phenomena is due to the fact that higher latent dimension increase the output vector size of NN and thereby reduce the NN's prediction accuracy on the latent vector. As was done for the number of FMs, by setting a threshold for $\text{Offset}_{mean}$ or $\text{Offset}_{max}$, the optimal latent dimension can be chosen accordingly for any geometric model. From the above results, we demonstrate that the proposed approach can be employed to predict the deformation of various soft tissue models. Moreover, the computation time for going through the entire prediction and reconstruction pipeline is less than 0.5 *s* for all the cases, indicating our work's potential for real-time surgical applications.

## Conclusion

A data-driven approach is developed for predicting intraoperative soft tissue deformation based on FM registration. In the proposed approach, physical nonlinear simulations are performed to model deformation prior to surgery and are later utilized to train a NN surrogate model for the real-time deployment of deformation prediction. We employ an AE encoder-decoder structure to project the full-length deformation vectors into a low-dimensional latent space, which simplifies the NN training and prediction process. For the H&N tumor model with a maximum displacement of 30 mm, our approach is capable of reconstructing this deformation with sub-millimeter accuracy, which is below the typical 1 mm resolution of interventional ultrasound. Parametric studies on the number of FMs, latent dimension, and NN structure are performed accordingly to characterize the optimal NN model structure. Further tests on different anatomical models have demonstrated the generalizability of our approach to various geometries and topologies.

Although generating datasets and training the NN can take several hours, the process can be done preoperatively. After the NN is trained, the deformation prediction of one case takes less than 0.5 s, compared to the computational time of around 5 minutes for each simulation via the FEM method. This demonstrates the proposed approach's capability in real-time tracking of intraoperative soft tissue deformations with high reconstruction fidelity. Overall, the presented approach is able to yield fast and accurate predictions on intraoperative soft tissue deformations, by harnessing the advantages from both high-fidelity simulations and the NN surrogate model, which demonstrate its potential of being clinically relevant in the future.

### Limitations & future work

Despite presenting a systematic study on the real-time reconstruction of soft tissue's deformation with an NN-based approach, our paper still has several limitations. Firstly, our approach requires a minimum of three fixed anchor points (boundary conditions) to restrict rotation of the model when applying force fields. When the fixed anchor points change, our NN-based model might not be able to yield accurate predictions and deformation reconstruction results. Secondly, it is well established that NNs perform poorly on out of range predictions. Therefore it is necessary to generate a deformation dataset which exceeds the in situ deformations for the surgical ROI. Additionally, the current approach requires that the to-be-reconstructed sample must be derived from the exactly same geometry and mesh topology. If the initial geometry of the sample is changed, the NN may fail to reconstruct its deformation.

Due to the aforementioned limitations, there is room for improvement to extend our current work. For instance: (i) we can generalize benchmark creation with more complex BCs, (ii) we can use physical experiments to validate our proposed computational framework, (iii) we can extend deformation to a larger scale with changes in topology, and (iv) we can employ more advance machine learning models/architectures.

## Supporting information

**S1 Appendix. Method for selecting FMs.** The conventional K-center clustering algorithm iteratively searches the farthest point from the pre-obtained centers within a specific closed geometry [33]. However, it might lead to an uneven distribution of search directions when the geometry is highly irregular, forcing the centers to be heavily concentrated at corners or edges of the region. Realistically, the geometries of human soft tissues are originally or deformed to be irregular in most cases. Therefore, to homogeneously distribute FMs in the target geometry, the revised K-center clustering algorithm maximizes the average and minimizes the variance of distances among the selected centers at the same time. In this paper, the minimal distance threshold between two center points $d_{min}$ is set to 10 mm; two sub-loops are implemented in the algorithm to alternately obtain the point with the maximum distance average and the point with the minimum distance variance. With an initial point specified, the algorithm can automatically search the next best center point candidate based on the positions of pre-obtained centers and iteratively distribute the required number of center points within a closed mesh topology. Algorithm S1 Fig shows the pseudocode detailing the implementation of the aforementioned algorithm. S1 Fig shows FMs captured by the algorithm with different numbers of iterations (i.e., number of points $k$) within the H&N tumor model, evidencing that the revised K-center clustering algorithm enables the homogeneous distribution of FMs with arbitrarily specified $k$.

**S1 Fig. Distribution of different numbers ($k$) of FMs in H&N tumor model.** Starting at the initial node index of 97, the red asterisks represent the centers obtained by the revised K-center clustering algorithm. $k = 5$ is eventually selected for implementations in this paper. (TIF)

## Acknowledgments

We thank Dr. Gal Shafirstein from the Photodynamic Therapy Center, Department of Cell Stress Biology, Roswell Park Comprehensive Cancer Center for providing H&N tumor geometry. We thank NIH 3D Exchange for providing us with the geometric models of the kidney [29] and aorta aneurysm [32].

## Author contributions

**Conceptualization:** Haolin Liu, Ye Han, Yoed Rabin, Levent Burak Kara.

**Formal analysis:** Haolin Liu, Ye Han, Daniel Emerson.

**Investigation:** Haolin Liu, Ye Han, Daniel Emerson.

**Methodology:** Haolin Liu, Ye Han, Daniel Emerson, Yoed Rabin, Levent Burak Kara.

**Project administration:** Yoed Rabin, Levent Burak Kara.

**Software:** Haolin Liu, Ye Han, Daniel Emerson.

**Supervision:** Yoed Rabin, Levent Burak Kara.

**Validation:** Haolin Liu.

**Visualization:** Ye Han, Daniel Emerson.

**Writing – original draft:** Haolin Liu, Ye Han, Daniel Emerson, Levent Burak Kara.

**Writing – review & editing:** Haolin Liu, Daniel Emerson, Yoed Rabin, Levent Burak Kara.

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
