## [Decision Letter · Decision Letter 0]

9 Sep 2024

PONE-D-24-17113A data-driven approach for real-time soft tissue deformation prediction using nonlinear presurgical simulationsPLOS ONE

Dear Dr. Kara,

Thank you for submitting your manuscript to PLOS ONE. After careful consideration, we feel that it has merit but does not fully meet PLOS ONE’s publication criteria as it currently stands. Therefore, we invite you to submit a revised version of the manuscript that addresses the points raised during the review process.

Please review and respond to the reviewer comments below. 

We look forward to receiving your revised manuscript.

Kind regards,

Joanna Tindall, PhD

Staff Editor

PLOS ONE

Journal Requirements: When submitting your revision, we need you to address these additional requirements. 1. Please ensure that your manuscript meets PLOS ONE's style requirements, including those for file naming. The PLOS ONE style templates can be found at https://journals.plos.org/plosone/s/file?id=wjVg/PLOSOne_formatting_sample_main_body.pdf and https://journals.plos.org/plosone/s/file?id=ba62/PLOSOne_formatting_sample_title_authors_affiliations.pdf 2. Please note that PLOS ONE has specific guidelines on code sharing for submissions in which author-generated code underpins the findings in the manuscript. In these cases, we expect all author-generated code to be made available without restrictions upon publication of the work. Please review our guidelines at https://journals.plos.org/plosone/s/materials-and-software-sharing#loc-sharing-code and ensure that your code is shared in a way that follows best practice and facilitates reproducibility and reuse. 3. When completing the data availability statement of the submission form, you indicated that you will make your data available on acceptance. We strongly recommend all authors decide on a data sharing plan before acceptance, as the process can be lengthy and hold up publication timelines. Please note that, though access restrictions are acceptable now, your entire data will need to be made freely accessible if your manuscript is accepted for publication. This policy applies to all data except where public deposition would breach compliance with the protocol approved by your research ethics board. If you are unable to adhere to our open data policy, please kindly revise your statement to explain your reasoning and we will seek the editor's input on an exemption. Please be assured that, once you have provided your new statement, the assessment of your exemption will not hold up the peer review process. 4. We notice that your supplementary tables are included in the manuscript file. Please remove them and upload them with the file type 'Supporting Information'. Please ensure that each Supporting Information file has a legend listed in the manuscript after the references list.

Reviewers' comments:

Reviewer's Responses to Questions

**Comments to the Author**

1. Is the manuscript technically sound, and do the data support the conclusions?

Reviewer #1: Yes

Reviewer #2: Yes

2. Has the statistical analysis been performed appropriately and rigorously? 

Reviewer #1: N/A

Reviewer #2: No

3. Have the authors made all data underlying the findings in their manuscript fully available?

Reviewer #1: No

Reviewer #2: No

4. Is the manuscript presented in an intelligible fashion and written in standard English?

Reviewer #1: Yes

Reviewer #2: Yes

5. Review Comments to the Author

Reviewer #1: This paper presents a methodology to predict in real-time the deformation of human tissues during image-guided surgeries. A dataset of FEM simulations under a range of force conditions is created with geometric models of a patient’s region of interest during surgery. The training dataset is employed to train an autoencoder, which develops a latent space representation capable of encoding tissue shape deformations. Subsequently, a neural network and a regression model are trained to link specific significant displacements to this latent space vectors of the deformation fields. The predicted latent space vectors are then decoded to obtain the deformation field vectors corresponding to the deformed tissue and predict in real time the deformed tissue. While the paper is globally well written and clear, there are some open issues which should be addressed before publication.

• Even if you choose non-linear material properties, the equation used for the FEM problem is for a linear problem. How do you explain this?

• Why do you use the same mechanical properties for the simulation of the three tissues? Do you have a reference to justify the parameters and all the properties in table 1?

• Line 92: “Deformations are primarily driven by the forces generated in the tissue’s interstitial environment”. Can you explain better the nature of these forces? Why are they generated? How? Where? Why do they influence the tissue deformation? Do you have some references for these forces?

• If Kin and Kout represent matrices they should be written in bold.

• Regarding the creation of the training and test datasets: Is the difference in the generated force field only in the magnitude and direction of the forces? Is the application point of the force always the same? If so, how do you choose the application point? The caption in figure 3 states: “randomly assigned node on the outer surface as the loading position,” but this is not explained in the main text.

• What is the computational time of the FEM simulations?

• Can you give more information about the variability of the dataset in the displacement field for each of the three cases tested? Is the variability in the range of clinically relevant data?

• You should provide more results for each example about the learning process of the NN. What is the behavior of the validation loss function with respect to the training loss function?

• Why do you use the same neural network architecture for the three examples?

• In the limitations, you state: “If the initial geometry of the sample is changed, the NN may fail to reconstruct its deformation.” Can you test the influence of the initial geometry on the results of your model? What is the clinical applicability of the model if we need to train another NN for each new geometry?

Reviewer #2: Summary of the work:

In this work, the authors present a ML surrogate model for FEM models for predicting tissue deformation given the displacements of some points (Fiducial Markers (FMs)). These FM are introduced during surgery procedures for real-time tracking of the region of interest (i.e., a tumor, a organ) through MRI or CT imaging, to then use then to track displacement with a lower fidelity method. With that patient-specific pretrained model, then a NN receives the displacements of the FM and transforms it to the latent space so the use the decoder of the VA to obtain the deformation of the region of interest.

Overall, I see the potential of the work and its application to clinical practice as it would help for more precise surgical interventions as it would allow a real-time representation of the region of interest with lower fidelity systems such as ultrasound imaging. The proposed data-driven approach makes sense as it allows taking advantage of the information from the FMs to perform real-time simulations and, in a further step, allow generalization across patients. However, I think there are several important concerns that should be addressed for publication, especially regarding the novelty of the work, the ML model, the information provided by the figures, and the flow of the paper.

Reviews

1. My first concern is regarding the novelty of the work. At lines 45-47 the authors state that existing literature rarely focuses on ML+FEM approaches. However, there are several works about combining FEM with ML to perform real-time deformation simulations. Just by doing a quick search I found:

Related works to the paper:

https://www.sciencedirect.com/science/article/pii/S0933365716304687

https://www.sciencedirect.com/science/article/pii/S0010482517303177

https://www.sciencedirect.com/science/article/pii/S0952197623013349

https://www.sciencedirect.com/science/article/pii/S0895611122001355

A review:

https://aapm.onlinelibrary.wiley.com/doi/full/10.1002/mp.14602

Additionally, some works that already presented FEM+ML to calculate organ deformations:

https://www.sciencedirect.com/science/article/pii/S0957417416306728

I would suggest that the authors should clarify in the introduction what they contribute compared to these works that employ similar approaches. In the same way that they review some works of FEM approaches (line 35), they should also mention the FEM+ML approaches and the benefits of the proposed work versus literature work.

2. I think that figure 1 is key to fully understand the procedure and needs improvement, especially regarding the flow of the figure and how the data is employed. As it is now, it’s a bit chaotic and not clear. Maybe it would benefit from dividing the figure with a. b. c… subfigures to make it clearer. In this way, the caption can explain in more details each part of the figure and then I think it may support the explanation in the manuscript in the material and methods section, as the text could reference each part of the figure as the manuscript explains the “creation of deformation…”, “encoding of deformation vectors”, “deformation prediction with NN”. Also, in the figures the during the surgery and before the surgery texts are not clear to where they span.

3. The authors show the methodology using head-and-neck (H&N) tumor and reference Figure 2, but in Figure 2 they also show other geometric models that are not commented. I recommend that they present those geometries and then establish that they will focus on H&N for explanation purposes, although I doubt about the value added by that figure.

4. In Figure 3 they show the force fields. Scale bar values are not clear. For each subfigure, n=1, n=3, and so on, what are the differences between those 3 geometries?

5. It does make sense to build an autoencoder to reduce the dimensionality of the problem and ease the deformation prediction. However, I have some doubts about the autoencoder model and training, there are few details about this. How is it trained? Is it the same dataset Gtrain, Gvalid, Gtest that is used to train the NN? In that case, wouldn’t the NN replicate the same latent space for which the AE was trained?

6. Line 163-165 parametric studies for the NN. However, could not find how they decided the optimal NN architecture of [n1, n2, n3] = [1024, 512, 256]. The parametric studies section is rather scarce, it does give little information.

7. I think it presenting the results and then presenting the parametric studies, number of FMs, and latent space does not help to the flow of the paper. I believe the paper would benefit from rearranging and reorganizing.

Minor things:

Reference 5, last accessed missing

Line 265 extra space in H& N

Conclusions:

I believe that the work needs some improvements to finally consider it for publication.

6. PLOS authors have the option to publish the peer review history of their article (what does this mean?). If published, this will include your full peer review and any attached files.

Reviewer #1: No

Reviewer #2: No

---

## [Author Response · Author response to Decision Letter 1]

13 Nov 2024

Revision overview: We thank both reviewers for their thoughtful comments and suggestions. We have replied to each comment below, and marked changes to the manuscript in red.

Reviewer #1: This paper presents a methodology to predict in real-time the deformation of human tissues during image-guided surgeries. A dataset of FEM simulations under a range of force conditions is created with geometric models of a patient’s region of interest during surgery. The training dataset is employed to train an autoencoder, which develops a latent space representation capable of encoding tissue shape deformations. Subsequently, a neural network and a regression model are trained to link specific significant displacements to these latent space vectors of the deformation fields. The predicted latent space vectors are then decoded to obtain the deformation field vectors corresponding to the deformed tissue and predict in real time the deformed tissue. While the paper is globally well written and clear, there are some open issues which should be addressed before publication.

Even if you choose non-linear material properties, the equation used for the FEM problem is for a linear problem. How do you explain this?

A: In our FEM simulation, the load on the external soft tissue surface, modeled as a force field boundary condition, was applied incrementally with a time history. The linear equation F=kx is simply a linear solver with the Newton-Raphson algorithm for one “incremental step” (to not be confused with numerical “solving steps”, we call it one “increment” hereafter). This means that despite simulating one increment linearly, the overall simulation history is considered nonlinear -- the initial deformed configuration and nodal variables of each increment are taken from the end frame of the previous simulation increment. Combined with material nonlinearity, we believe this simulates nonlinear behaviors of soft tissue deformation.

We have made the following change at Line 130 in the manuscript to better explain the nonlinearity of our simulation pipeline: “In essence, the nonlinearity of our simulation arises from both nonlinear material properties and nonlinear simulation schemes. This is sufficient to capture most of the nonlinear deformation behaviors of the target soft tissues. ”.

Why do you use the same mechanical properties for the simulation of the three tissues? Do you have a reference to justify the parameters and all the properties in Table 1?

A: Thanks for the reviewer’s suggestion. Our main goal is to show the efficacy of our framework regarding predicting full-field deformation given fiducial markers’ displacements. To make the corresponding studies comparable in terms of overall displacement magnitudes, we kept the materials properties the same across the different examples we present. Since we generated latent distribution based on material-property-agnostic deformation data, we believe that the machine learning model would yield equally accurate results with whatever material properties we choose as long as the soft tissue deformation magnitude falls within a similar range.

However, we do acknowledge the necessity to demonstrate the aforementioned point, therefore we identified from the literature the material properties of kidney and aorta tissue and changed them accordingly in the simulation model. We find that the kidney has been modeled as a neo-Hookean solid with an elastic modulus of 180 kPa, while the aorta tissue has been modeled as a neo-Hookean solid with an elastic modulus of 250 kPa. Note that we too model these two tissues as neo-Hookean solids.

Table 1. Simulation results of average nodal deformation (mm) of kidney and aorta soft tissue models with newly found realistic material properties.

Average nodal deformation (mm)

Example No.

1

2

3

4

5

Kidney model with new properties

11.35

24.18

6.35

3.77

7.72

Aorta model with new properties

3.39

4.27

5.49

1.98

4.16

We performed batch simulations for the kidney (in a total of 20 simulations) and aorta (in a total of 30 simulations) models with the newly found material properties from literature and similar boundary conditions, and reported 5 of each model in the above table. We found that after performing the same simulation procedures, the average of the average nodal deformation for the kidney and aorta is 12.46 mm and 4.73 mm, respectively. This falls within the range of what we have simulated with the material properties of the head and neck tumor, therefore we believe this is sufficient to demonstrate that our data-driven framework will work well with these two new material properties and similar load conditions.

We have made the following change at Line 97 in the manuscript to better explain the big picture of our idea and resolve this issue: “It is worth noting that we used the same material properties from the H&N tumor for our kidney and aorta aneurysm models as well, since the major aim of this research is to show the capability of our data-driven pipeline on full-field deformation prediction given only a few displacements values of FMs, rather than generating realistic deformation fields for each respective model. We hypothesize that our trained machine learning model would yield equally accurate results with whatever material properties we choose as long as the soft tissue deformation magnitude falls within a similar range. ”.

We list the reference for the aforementioned kidney and aorta tissue material properties below:

Aorta tissue: Okamoto, Ruth J., et al. "Mechanical properties of dilated human ascending aorta." Annals of biomedical engineering 30 (2002): 624-635.

Kidney: Karimi, A., and A. Shojaei. "Measurement of the mechanical properties of the human kidney." Irbm 38.5 (2017): 292-297.

Line 92: “Deformations are primarily driven by the forces generated in the tissue’s interstitial environment”. Can you explain better the nature of these forces? Why are they generated? How? Where? Why do they influence tissue deformation? Do you have some references for these forces?

A: We generated the force fields by (a) randomly creating up to three nodal forces on the surface of the tissue, and (b) applying the Laplace-Beltrami operator to smooth the aforementioned nodal forces to the entire tissue surface. The purpose of force field generation on the tissue surface is to create smooth yet non-homogeneous distributed loads (force fields) that mimic a realistic environment a tissue is embedded in, where the surface contact forces applied to the tissue could potentially be from arbitrary directions with magnitudes causing mild deformation. To make sure we create a dataset with a large variety of force fields based on the above conditions, we randomly pick up to three surface nodes and sample their magnitudes in either a uniform or a Gaussian way.

The force field smoothing has been re-depicted with a newly created Figure 3. And we have made the following change in Line 138 of the manuscript to explain our force field generation better: “As shown in Figure. 4, we first generate a few nodal forces on the surface of the model by randomly selecting positions, magnitudes, and directions of the loads; then we perform spectral decomposition of the discretized Laplacian of the tumor's surface mesh to obtain a discretized Laplacian-Beltrami operator (LBO) that can smoothly spread the nodal forces into smoother force field distributions around them.”.

If Kin and Kout represent matrices they should be written in bold.

A: We thank the reviewer for the good catch! We have made Kin and Kout bold to indicate that they are matrices. The corrected sentence is now red in LIne 112.

Regarding the creation of the training and test datasets: Is the difference in the generated force field only in the magnitude and direction of the forces? Is the application point of the force always the same? If so, how do you choose the application point? The caption in figure 3 states: “randomly assigned node on the outer surface as the loading position,” but this is not explained in the main text.

A: As we mentioned in the previous question, we randomly picked the load positions, directions, and magnitudes to try to create a diverse enough deformation dataset. We also want to clarify that as we apply our system, we do not need to input the location of the realistic load -- the only input should be the geometry of the model, the location of fiducial markers, and fiducial marker displacements. As such, our data-driven framework is load-condition-agnostic and would perform better if the training dataset had richer selections of loads.

What is the computational time of the FEM simulations?

A: We thank the reviewer for the question. The simulation time of our FEM is on average 5 mins, given specific boundary conditions on the surface of the tissue model. In a real intraoperative application, however, it could be challenging to obtain and keep tracking accurate load conditions on the outer surface of the tissue. Therefore, the actual time could be much longer given that more time is needed to accurately model the boundary conditions. We have added the following sentence in the Conclusion section of the manuscript to highlight the computational time advantage of our framework: “After the NN is trained, the deformation prediction of one case takes less than 0.5 s, compared to the computational time of around 5 mins for each simulation via the FEM method. This demonstrates the proposed approach’s capability in real-time tracking of intraoperative soft tissue deformations with high reconstruction fidelity.”.

Can you give more information about the variability of the dataset in the displacement field for each of the three cases tested?

A: The following table shows the variability of our simulation results used to generate our training and testing dataset.

H&N tumor

Kidney

Aorta

Max. nodal deformation (mm)

18.33

25.72

15.74

Min. nodal deformation (mm)

0.79

0.45

0.34

The standard deviation of nodal deformation (mm)

3.87

6.66

2.14

Is the variability in the range of clinically relevant data?

A: Yes, the variability in our dataset is within the range of clinically relevant data. While it is challenging to find exact reference values for each specific geometry, several papers cite the range of nodal deformations during image guided liver surgery as between 5 to 20mm (Clements et al. 2011 & 2017), as well as aforementioned deformation modeling papers which simulated displacements up to 15mm (Lorente et al. 2017).

Clements, Logan W., et al. "Deformation correction for image guided liver surgery: An intraoperative fidelity assessment." Surgery 162.3 (2017): 537-547.

Clements, Logan W., et al. "Organ surface deformation measurement and analysis in open hepatic surgery: method and preliminary results from 12 clinical cases." IEEE transactions on biomedical engineering 58.8 (2011): 2280-2289.

Lorente, Delia, et al. "A framework for modelling the biomechanical behaviour of the human liver during breathing in real time using machine learning." Expert Systems with Applications 71 (2017): 342-357.

You should provide more results for each example about the learning process of the NN. What is the behavior of the validation loss function with respect to the training loss function?

A: We monitor the loss curves during training to ensure there is not an increase in the validation curve while the training curve continues decreasing - this would indicate overfitting to the training data. We briefly address this point in Sec. Deformation Prediction with NN, but feel it does not warrant new figures as this concept is well established within the field of machine learning and may be either superfluous information or potentially confusing information depending on the reader’s background.

Why do you use the same neural network architecture for the three examples?

A: Good question - we tested out many different hidden layer structures and found empirically that the [1024, 512, 256] structure appropriately balanced generalizability without a strong propensity to overfit to training data across all models. Rather than fine-tuning the model to obtain marginally better results for each geometry, we present a single architecture that performs well across all patient-specific geometries. We have added several sentences to clarify this point in Sec. Deformation Prediction with NN.

In the limitations, you state: “If the initial geometry of the sample is changed, the NN may fail to reconstruct its deformation.” Can you test the influence of the initial geometry on the results of your model? What is the clinical applicability of the model if we need to train another NN for each new geometry?

A: Thank you for the question - neural networks typically struggle with out-of-bounds prediction, meaning that a model trained on deformations for a specific geometry will struggle to generate quality predictions on a new, unseen geometry. For example - this would be like taking a model trained on patient A’s H&N tumor, and then using that trained model to guide the surgery on patient B’s tumor. It is much more clinically appropriate to train a new model based on patient specific geometries and deformation to ensure the highest degree of accuracy when using the model to guide a surgery. That said, we train the models on a diverse range of deformations for a specific geometry to ensure the model can accurately predict a broad range of deformations that might be seen during surgery.

Reviewer #2: Summary of the work:

In this work, the authors present a ML surrogate model for FEM models for predicting tissue deformation given the displacements of some points (Fiducial Markers (FMs)). These FM are introduced during surgery procedures for real-time tracking of the region of interest (i.e., a tumor, a organ) through MRI or CT imaging, to then use then to track displacement with a lower fidelity method. With that patient-specific pretrained model, then a NN receives the displacements of the FM and transforms it to the latent space so the use the decoder of the VA to obtain the deformation of the region of interest.

Overall, I see the potential of the work and its application to clinical practice as it would help for more precise surgical interventions as it would allow a real-time representation of the region of interest with lower fidelity systems such as ultrasound imaging. The proposed data-driven approach makes sense as it allows taking advantage of the information from the FMs to perform real-time simulations and, in a further step, allow generalization across patients. However, I think there are several important concerns that should be addressed for publication, especially regarding the novelty of the work, the ML model, the information provided by the figures, and the flow of the paper.

My first concern is regarding the novelty of the work. At lines 45-47 the authors state that existing literature rarely focuses on ML+FEM approaches. However, there are several works about combining FEM with ML to perform real-time deformation simulations. Just by doing a quick search I found:

Related works to the paper:

https://www.sciencedirect.com/science/article/pii/S0933365716304687

https://www.sciencedirect.com/science/article/pii/S0010482517303177

https://www.sciencedirect.com/science/article/pii/S0952197623013349

https://www.sciencedirect.com/science/article/pii/S0895611122001355

A review:

https://aapm.onlinelibrary.wiley.com/doi/full/10.1002/mp.14602

Additionally, some works that already presented FEM+ML to calculate organ deformations:

https://www.sciencedirect.com/science/article/pii/S0957417416306728

I would suggest that the authors should clarify in the introduction what they contribute compared to these works that employ similar approaches. In the same way that they review some works of FEM approaches (line 35), they should also mention the FEM+ML approaches and the benefits of the proposed work versus literature work.

A: Great point, thank you for pointing us towards all of these prior works. We feel that our paper is still well situated within the space, and have added a paragraph discussing these works, and make the case that our fiducial marker based approach makes use of an already common metho

---

## [Decision Letter · Decision Letter 1]

30 Dec 2024

PONE-D-24-17113R1A data-driven approach for real-time soft tissue deformation prediction using nonlinear presurgical simulationsPLOS ONE

Dear Dr. Kara,

Thank you for submitting your manuscript to PLOS ONE. After careful consideration, we feel that it has merit but does not fully meet PLOS ONE’s publication criteria as it currently stands. Therefore, we invite you to submit a revised version of the manuscript that addresses the points raised during the review process.

**The Reviewers queried about the simulation details, e.g., the dataset, the adoption of same Young's Modulus for different organs, and the risk of information loss during compression. Explanation for the data-driven method also lacked. Relevant explanation and descriptions should be given in the revised version.**

We look forward to receiving your revised manuscript.

Kind regards,

Longhui Qin, Ph.D.

Academic Editor

PLOS ONE

Reviewers' comments:

Reviewer's Responses to Questions

**Comments to the Author**

1. If the authors have adequately addressed your comments raised in a previous round of review and you feel that this manuscript is now acceptable for publication, you may indicate that here to bypass the “Comments to the Author” section, enter your conflict of interest statement in the “Confidential to Editor” section, and submit your "Accept" recommendation.

Reviewer #1: All comments have been addressed

Reviewer #3: All comments have been addressed

2. Is the manuscript technically sound, and do the data support the conclusions?

Reviewer #1: Yes

Reviewer #3: Partly

3. Has the statistical analysis been performed appropriately and rigorously? 

Reviewer #1: N/A

Reviewer #3: Yes

4. Have the authors made all data underlying the findings in their manuscript fully available?

Reviewer #1: Yes

Reviewer #3: Yes

5. Is the manuscript presented in an intelligible fashion and written in standard English?

Reviewer #1: Yes

Reviewer #3: Yes

6. Review Comments to the Author

**Reviewer #1:** The authors have adequately addressed the suggestions made by the reviewers, modifying the original manuscript.

Minor Comment: In equations 1 and 2, the terms corresponding to matrices and vectors should be in bold.

**Reviewer #3: **This paper proposes a data-driven method for real-time prediction of soft tissue deformation, aiming to address the shape mismatch between high-resolution preoperative images and low-resolution intraoperative images. The study utilizes the finite element method (FEM) to generate an offline deformation dataset, compresses the high-dimensional deformation field into a low-dimensional latent space via an autoencoder, and trains a neural network (NN) to map the displacement of landmarks to the latent space representation. Finally, the 3D deformation shape is reconstructed through a decoder. The method was tested on three geometric models—head and neck tumors, kidneys, and aortic aneurysms—demonstrating sub-millimeter prediction accuracy and real-time computational performance. However, several issues are noted:

1. How is the reliability of the dataset established? Variations in boundary conditions and mesh resolutions in FEM simulations can have a significant impact on the results. However, the paper does not provide evidence to confirm that the results are independent of these factors.

2. Are the references cited in 10–22 really necessary? This section seems to have an unusually large number of citations, are they all essential to support the argument?

3. The paper is divided into too many small sections, which affects the overall readability. It might be more effective to group the content into three main parts: background introduction, method development, and result discussion, with clear summary paragraphs for each section to enhance clarity.

4. Why do the head and neck, kidney, and aortic aneurysm models all share the same Young’s modulus? The deformation dataset appears to rely on fixed material properties, but there is no validation of how variations in material properties, such as changes in elastic modulus, might influence the results.

5. How were the parameters for the NN’s hidden layers determined? The paper does not provide sufficient explanation or a clear rationale for the choice of the NN’s loss function.

6. Would this method need to be retrained for patients with different constitutions or tissue characteristics? This aspect has not been addressed in the paper.

7. When compressing the deformation field into a low-dimensional latent space, is there a risk of losing important deformation information? The paper does not discuss this potential concern.

8. Are the test data generated in the study comparable in complexity to real intraoperative data?

7. PLOS authors have the option to publish the peer review history of their article (what does this mean?). If published, this will include your full peer review and any attached files.

Reviewer #1: No

Reviewer #3: No

---

## [Author Response · Author response to Decision Letter 2]

17 Jan 2025

Revision overview: Thank you to both of the reviewers for their insightful comments and

questions. We have replied to each point below and marked changes to the manuscript in red

text.

Reviewer #1: The authors have adequately addressed the suggestions made by the reviewers,

modifying the original manuscript.

1.) Minor Comment: In equations 1 and 2, the terms corresponding to matrices and vectors

should be in bold.

A: Thank you for pointing this out, we have bolded the matrices and vectors in these

equations.

Reviewer #3: This paper proposes a data-driven method for real-time prediction of soft tissue

deformation, aiming to address the shape mismatch between high-resolution preoperative

images and low-resolution intraoperative images. The study utilizes the finite element method

(FEM) to generate an offline deformation dataset, compresses the high-dimensional

deformation field into a low-dimensional latent space via an autoencoder, and trains a neural

network (NN) to map the displacement of landmarks to the latent space representation. Finally,

the 3D deformation shape is reconstructed through a decoder. The method was tested on three

geometric models—head and neck tumors, kidneys, and aortic aneurysms—demonstrating

sub-millimeter prediction accuracy and real-time computational performance. However, several

issues are noted:

1.) How is the reliability of the dataset established? Variations in boundary conditions

and mesh resolutions in FEM simulations can have a significant impact on the

results. However, the paper does not provide evidence to confirm that the results

are independent of these factors.

A: For force field boundary conditions, we generated them by (a) randomly

creating up to three nodal forces on the surface of the tissue, and (b) applying the

Laplace-Beltrami operator to smooth the aforementioned nodal forces to the entire

tissue surface. The purpose of force field generation on the tissue surface is to

create smooth yet non-homogeneous distributed loads (force fields) that mimic a

realistic environment a tissue is embedded in, where the surface contact forces

applied to the tissue could potentially be from arbitrary directions with magnitudes

causing mild deformation. To make sure we create a dataset with a large variety of

force field boundary conditions based on the above conditions, we randomly pick

up to three surface nodes and sample their magnitudes in either a uniform or a

Gaussian way. The force fields are then smoothed using the Laplace-Beltrami

operator, the process of which is depicted in Figure 3.

Since all of our simulations converged successfully, we believe that mesh

resolution would not change the results drastically. Therefore, it is not necessary to

change mesh resolutions and it won’t change the essential conclusion of our

paper.

2.) Are the references cited in 10–22 really necessary? This section seems to have

an unusually large number of citations, are they all essential to support the

argument?

A: We have removed the citations which are not directly related to soft tissue

deformation prediction, namely citations [11] Lowe et al. and [22] Jaradat et al.

3.) The paper is divided into too many small sections, which affects the overall

readability. It might be more effective to group the content into three main parts:

background introduction, method development, and result discussion, with clear

summary paragraphs for each section to enhance clarity.

A: Thank you for the suggestion. The paper is already primarily organized with the

main sections that you suggest; introduction, methods, results/discussion. We

further break these sections into subsections and subsubsections which are titled

in a smaller font. We feel this structure offers better clarity to the reader rather than

simply having a large wall of text under each section with the only distinction being

paragraph breaks.

4.) Why do the head and neck, kidney, and aortic aneurysm models all share the

same Young’s modulus? The deformation dataset appears to rely on fixed material

properties, but there is no validation of how variations in material properties, such

as changes in elastic modulus, might influence the results.

A: Our main goal is to show the efficacy of our framework regarding predicting

full-field deformation given fiducial markers’ displacements. To make the

corresponding studies comparable in terms of overall displacement magnitudes,

we kept the material properties the same across the different examples we

present. Since we generated latent distribution based on

material-property-agnostic deformation data, we believe that the machine learning

model would yield equally accurate results with whatever material properties we

choose as long as the soft tissue deformation magnitude falls within a similar

range.

However, we do acknowledge the necessity to demonstrate the aforementioned

point, therefore we identified from the literature the material properties of kidney

and aorta tissue and changed them accordingly in the simulation model. We find

that the kidney has been modeled as a neo-Hookean solid with an elastic modulus

of 180 kPa, while the aorta tissue has been modeled as a neo-Hookean solid with

an elastic modulus of 250 kPa. Note that we too model these two tissues as

neo-Hookean solids.

Table 1. Simulation results of average nodal deformation (mm) of kidney and aorta

soft tissue models with newly found realistic material properties.

Average nodal deformation (mm)

Example No. 1 2 3 4 5

Kidney model with

new properties

11.35 24.18 6.35 3.77 7.72

Aorta model with

new properties

3.39 4.27 5.49 1.98 4.16

We performed batch simulations for the kidney (in a total of 20 simulations) and

aorta (in a total of 30 simulations) models with the newly found material properties

from literature and similar boundary conditions, and reported 5 of each model in

the above table. We found that after performing the same simulation procedures,

the average of the average nodal deformation for the kidney and aorta is 12.46

mm and 4.73 mm, respectively. This falls within the range of what we have

simulated with the material properties of the head and neck tumor, therefore we

believe this is sufficient to demonstrate that our data-driven framework will work

well with these two new material properties and similar load conditions.

In the “Materials and Methods” section of the paper, we explain this point as

follows: “It is worth noting that we used the same material properties from the H&N

tumor for our kidney and aorta aneurysm models as well, since the major aim of

this research is to show the capability of our data-driven pipeline on full-field

deformation prediction given only a few displacements values of FMs, rather than

generating realistic deformation fields for each respective model. We hypothesize

that our trained machine learning model would yield equally accurate results with

whatever material properties we choose as long as the soft tissue deformation

magnitude falls within a similar range.

”.

We list the reference for the aforementioned kidney and aorta tissue material

properties below:

Aorta tissue: Okamoto, Ruth J., et al.

"Mechanical properties of dilated human

ascending aorta.

" Annals of biomedical engineering 30 (2002): 624-635.

Kidney: Karimi, A., and A. Shojaei.

"Measurement of the mechanical properties

of the human kidney.

" Irbm 38.5 (2017): 292-297.

5.) How were the parameters for the NN’s hidden layers determined? The paper does

not provide sufficient explanation or a clear rationale for the choice of the NN’s

loss function.

A: For the NN we used, we tried to make the fully connected structure as simple

as possible since it mainly works on mapping between two vectors. We performed

test runs over a few three-layer multilayer perceptrons with different numbers of

hidden layer neurons, and eventually fixed the structure to the current

configuration. For the loss function, we chose to use the mean square error loss

(MSELoss) since it is one of the most widely used ones and fits the task we

designed in this research.

6.) Would this method need to be retrained for patients with different constitutions or

tissue characteristics? This aspect has not been addressed in the paper.

A: Yes, the model should be patient-specific (as stated in the last paragraph of the

Introduction section) and needs to be retrained for different patients or different

tissue properties. However, if it can be experimentally determined that the material

properties and shape/structure of tissues (e.g., livers) of two patients are almost

identical, the same model could be used to directly guide the intraoperative

deformation tracking during surgeries. However, we believe that the main selling

point of our paper is to show the capability and efficacy of our proposed framework

- a framework that leverages machine learning techniques and fiducial marker

displacements to reconstruct the full-field deformation of the tissue. This

framework allows surgeons to simulate & train models with given patient-specific

properties and parameters before the surgery, which is easy & cost-effective to

achieve and can bring benefits to surgery monitoring.

7.) When compressing the deformation field into a low-dimensional latent space, is

there a risk of losing important deformation information? The paper does not

discuss this potential concern.

A: Good question, we address this point in the Parametric Studies: Latent

Dimension section. We perform a parametric sweep of latent dimensions and

select the dimension according to the “elbow method,

” where higher dimensions

offer diminishing returns in nodal reconstruction error. We further discuss how the

selection of the latent dimension also must consider the added computational cost

as well as the interaction with the neural network which maps between the fiducial

marker displacements and the latent vector.

8.) Are the test data generated in the study comparable in complexity to real

intraoperative data?

A: Yes, the test data is comparable in complexity to real intraoperative data both in

terms of the scale of deformations, complexity of the geometry, and the number of

fiducial markers used.

The following table shows the variability of our simulation results used to generate

our training and testing dataset.

H&N tumor Kidney Aorta

Max. nodal deformation (mm) 18.33 25.72 15.74

Min. nodal deformation (mm) 0.79 0.45 0.34

The standard deviation of

nodal deformation (mm) 3.87 6.66 2.14

While it is challenging to find exact reference values for each specific geometry,

several papers cite the range of nodal deformations during image guided liver

surgery as between 5 to 20mm (Clements et al. 2011 & 2017), as well as

aforementioned deformation modeling papers which simulated displacements up

to 15mm (Lorente et al. 2017).

Furthermore, we evaluate our framework’s performance on the three geometries;

H&N tumor, kidney, and aorta aneurysm to add additional geometric complexity.

The kidney poses an additional challenge due to its hollow structure, and the aorta

due to its hollow structure and thin walls.

With regards to the complexity of fiducial marker displacements used in our

framework, the H&N tumor models use just 5 fiducial markers, and the aorta

aneurysm uses 10 fiducial markers. Clinically, a minimum of 3 fiducial markers

must be implanted. That said, the 10 fiducial markers used in the aorta aneurysm

case may exceed what is practical. This is an important clinical consideration,

because fiducial markers must be surgically implanted prior to the surgery. For all

three cases, we perform a parametric study to select the minimum number of

fiducial markers which yields deformation reconstruction error which is below a

threshold.

Clements, Logan W., et al.

"Deformation correction for image guided liver surgery: An

intraoperative fidelity assessment.

" Surgery 162.3 (2017): 537-547.

Clements, Logan W., et al.

"Organ surface deformation measurement and analysis in open

hepatic surgery: method and preliminary results from 12 clinical cases.

" IEEE transactions

on biomedical engineering 58.8 (2011): 2280-2289.

Lorente, Delia, et al.

"A framework for modelling the biomechanical behaviour of the

human liver during breathing in real time using machine learning.

" Expert Systems with

Applications 71 (2017): 342-357.

---

## [Decision Letter · Decision Letter 2]

29 Jan 2025

A data-driven approach for real-time soft tissue deformation prediction using nonlinear presurgical simulations

PONE-D-24-17113R2

Dear Dr. Kara,

We’re pleased to inform you that your manuscript has been judged scientifically suitable for publication and will be formally accepted for publication once it meets all outstanding technical requirements.

Kind regards,

Longhui Qin, Ph.D.

Academic Editor

PLOS ONE

Additional Editor Comments (optional):

Reviewers' comments:

Reviewer's Responses to Questions

**Comments to the Author**

1. If the authors have adequately addressed your comments raised in a previous round of review and you feel that this manuscript is now acceptable for publication, you may indicate that here to bypass the “Comments to the Author” section, enter your conflict of interest statement in the “Confidential to Editor” section, and submit your "Accept" recommendation.

Reviewer #3: All comments have been addressed

2. Is the manuscript technically sound, and do the data support the conclusions?

Reviewer #3: Yes

3. Has the statistical analysis been performed appropriately and rigorously? 

Reviewer #3: Yes

4. Have the authors made all data underlying the findings in their manuscript fully available?

Reviewer #3: Yes

5. Is the manuscript presented in an intelligible fashion and written in standard English?

Reviewer #3: Yes

6. Review Comments to the Author

Reviewer #3: (No Response)

7. PLOS authors have the option to publish the peer review history of their article (what does this mean?). If published, this will include your full peer review and any attached files.

Reviewer #3: No

---

## [Editor Report · Acceptance letter]

PONE-D-24-17113R2

PLOS ONE

Dear Dr. Kara,

I'm pleased to inform you that your manuscript has been deemed suitable for publication in PLOS ONE. Congratulations! Your manuscript is now being handed over to our production team.

Kind regards,

on behalf of

Prof. Longhui Qin

Academic Editor

PLOS ONE